# COVID-19 and Vulnerable Children Well-Being: Interview with Left-Behind Children in Rural China

**DOI:** 10.3390/children9091317

**Published:** 2022-08-29

**Authors:** Endale Tadesse, Sabika Khalid, Cai Lianyu, Chunhai Gao

**Affiliations:** 1College of Teacher Education, Zhejiang Normal University, Jinhua 321017, China; 2Faculty of Education, Shenzhen University, Shenzhen 518060, China

**Keywords:** COVID-19, left-behind children, psychological adjustment, academic adjustment, parental attachment, well-being

## Abstract

Purpose of the study: This study sought to explore the psychological well-being, academic adjustment, and quality of parental attachment of LBC during COVID-19 based on Left-Behind Children’s (LBC) word of mouth. Method: In light of the abundance of quantitative studies, this qualitative study explored the psychological, academic, and parental attachment experiences of rural LBC during COVID-19. To this end, we conducted semi-structured interviews with 22 LBCs aged 10–15 in May 2021. Result: The study results reveal that LBCs developed severe psychological illnesses after the pandemic severely disrupted their already disturbed lives. Our findings showed that most LBCs feel they do not need their parents, which reflects their long record of hopelessness and agony regarding the absence of their parents. Conclusion: Although COVID-19 is a global problem, its impact was particularly devastating for LBCs who have nobody around who could help them with their academic, personal and social need. In the modern COVID-19 era, it seems that Chinese grandparenting has become unreliable in the face of radical change in contemporary Education, society, and the economic system. Therefore, this study suggests that the Chinese government should seek to identify and monitor these children by working with NGOs that target such children.

## 1. Introduction

The recent rural-to-urban migration in China for radical economic growth and development has brought massive migration that results in most parents leaving their children under the care of their patriarchs, families, or legitimate custodians [1,2,3,4,5]. As the National Bureau of Statistics (2016) reported, in 2015 more than 280 million workers moved from the rural side of China to the urban side, which resulted in unbalanced urbanization [6,7]. In order to control internal peasant migration and keep balance in the country, China’s government launched the hukou system in 1958 (i.e., the household registration system) [6,7,8,9]. Later, however, in the 1970s, relaxation in the hukou system for rapid economic development and the need for labor forced encouraged rural workers to migrate to cities [10]. As per the hukou, the birth location or household hukou registration district is the destination where migrant parents’ children can receive free basic education (class level 1–9). To attend a city school, migrants’ parents have to pay a heavy enrollment fee that causes stress [10,11,12]. This forced migrant parents to leave their children with a family member in rural areas to pursue free basic education for their children [9]. This study refers to these children as left-behind children (LBC). Under the age of 18, one-in-three children is a left-behind child (LBC) whose parents or one parent has migrated [10]. A big-large study based on the national population census data claimed that almost half of the members in the research were children left behind by one or both migrant parents [2]. Conferring to current studies, the status of rural children whose parents migrate can be classified into three groups: (a) children who migrate to cities with their parents; (b) partially LBC, who settle in rural areas with one parent who did not migrate; and (c) completely LBC, who remain with their grandparents in their home village after both parents have migrated to a city [13,14,15]. These three categories of children represent the most vulnerable children and adolescents; nevertheless, LBCs who remain behind in rural villages with one or none of their parents due to their migrant parents’ financial constraints or low income face psychological, mental, and academic challenges [16].

Similarly, a large body of Chinese evidence reveals that parental absence jeopardizes LBC food security and nutrition [17], academic performance [11], mental health [18], child development [10,19], and parental support [7,18]. Like this, many children are left behind (LBC) in rural areas under the supervision of caregivers, typically grandparents and relatives, when one or both parents migrate to cities [20]. Unfortunately, there is a high likelihood that these children live with elderly, uncaring, or abusive caregivers. Parents’ early-age parting of LBC causes significant psychological and mental issues [21,22,23]. Moreover, due to the intensive working hours and job engagement of migrants, they cannot visit their LBC [24], which causes these children to experience loneliness, a lack of communication, and sadness [21] due to insufficient family care and parental love. The current study sought to understand the experiences of rural LBCs in China, who are the most disadvantaged and confounded group of children during COVID-19 [2,25]. The study explored how this certain group of children experienced COVID-19 and how their parental support and caregivers helped them sustain their academic, psychological and parental attachment.

### 1.1. Literature Review

#### 1.1.1. Psychological Challenges of LBC

Previous studies have examined the psychological and mental well-being of LBCs in view of their parental absence [7,21,23,25,26]. Similarly, substantial evidence has affirmed that LBCs are at high psychological risk due to parental separation [7,23,27,28]. Although the caregivers of LBC (relatives or grandparents) strive to compensate for their threatened emotional attachment to their migrant parents by spending time with them or encouraging them to see their peers, such behavior cannot substitute or compensate for the lack of an actual parental physical and emotional presence [1,28]. Although migrant parents endeavor to maintain genuine communication with their LBC via social media, the odds of successfully sustaining the children’s psychological and mental well-being are notably narrow [11,28,29]. During early and middle adolescence, children require intensive physical involvement, support, and warmth from their parents to adapt appropriately to their environment. Therefore, video or voice communication between migrant parents and LBC alone is insufficient to guide children to develop the psychological skills needed to overcome problems or support their children in their daily challenges, and it deters physical bonding. A recent study plausibly argued that migrant parents could minimize the risk of LBC psychological and mental problems by engaging in solid communication with the caregivers (relatives or grandparents) who physically accompany LBC [28]. Unfortunately, because the traditional supervision by the elderly grandparents who primarily look after LBC is poor, the children can become involved in harmful activities, such as child labor, substance abuse, crime, and illegal behavior, and some have even committed suicide [10,21,30,31,32]. A book by Mu and Hu depicts the alarming reality of LBC maltreatment by caregivers in the form of nine tragic LBC stories that relate to the emergence of personality disorders, suicide due to anxiety, and death due to unsafe labor [22]. In addition, a meta-analysis states that nearly one-third of LBCs in China encounter physical and mental abuse by the legal guardians who care for them [33].

#### 1.1.2. Academic Challenges of LBC

The literature claimed that lower socioeconomic status (SES) urged parents to migrate towards urban areas, influencing children’s intellectual and non-intellectual capabilities [7,10,34,35]. The literature claims that early childhood academic provisional predicts the future educational outcomes of children [1,5,7]. Since migrants send remittances to their children, it does not ensure the significant amount that can provide the educational resources and support emotional connection [7,36,37]. An interview study in Sichuan Province with LBCs claimed that most LBCs are deprived of access to quality food, hygiene facilities, and parental attachment development. The latter failure often leads migrant parents to lose interest in their children’s education [4].

Furthermore, an emerging line of research in China has demonstrated that the unfavorable effect of parental migration on LBC academic adjustment is more profound for LBCs from a household where only one parent migrates to cities or other provinces, as opposed to both parents migrating [10,12]. A family in which only one parent migrates promotes a more disadvantaged academic environment for an LBC, as the parent who remains home requires the child or children to help with household or agricultural chores, and the remittance sent by the migrant parent is insufficient to cover the child’s academic resource needs; specifically, maternal migration makes things worse. Mother’s absence frequently results in LBCs being assigned household responsibilities and other day-to-day labor that consumes their study time [25,37], negatively affecting academic performance [8]. In contrast, LBCs who have both parents migrate may possess a moderate academic advantage compared to LBCs with one parent remaining at home while the other parent migrates, as two migrant parents together are more likely to be able to send a remittance sufficient for household needs and LBC educational support, such as private tutoring and extracurricular activities [5,32]. Interestingly, many recent studies argue that most LBCs in China appreciate their migrant parents’ efforts and try to perform well academically with their counterparts [11,18].

### 1.2. Parent-Child Relationship and Attachment among LBCs

In China, parent-child attachment is essential for children’s academic, social, physical, and psychological well-being [3,35,38]. The family is the most substantial environment in which children receive a legitimate experience to foster their relationship with their parents through academic and non-academic discussion [34]. The psychological stability of LBCs can predict their parent-child relationship, which can also determine their academic performance [39]. The previously mentioned findings notwithstanding, recent Chinese research has demonstrated that educational achievement may not be an issue if parental relationships and support are presently being a left-behind child [18]. Nevertheless, most migrants fail to maintain a robust parent-child relationship [11,15,39]. Thus, parents’ actual and emotional absence causes their children to exhibit adverse behaviors, such as denial of participating in social and educational activities [33,36]. While regular financial support from migrant parents for their children to live a quality life in rural areas is not possible unless these migrant parents can establish a firm interaction and communication with their LBC, there is a probability that the children will display and exhibit their motivation in their academic development [5,32].

### 1.3. Grandparent Caregiving and LBC

According to a recent study, 81% of LBCs in China are under the care of grandparents [40], which underlines the continuous massive interprovincial migration of both parents from one household while leaving behind their children with their grandparents, which is common in Chinese rural society [2,4,5,19,20]. The value of grandparents in overall Chinese family functionality is noteworthy. Rural LBC grandparents have shown a decade of solid attachment and assurances with LBC, which can perhaps compensate for the physical breach between migrant parents and LBC, and reduce mental stress and depression. On the one hand, an LBC from a household from which both parents have migrated may find a silver lining, so to speak, in this situation, by obtaining a more substantial income than a household in which only one or no parent has migrated, which allows LBC to satisfy necessary needs and enhance their life satisfaction. On the other hand, LBCs display low social competency and experience loneliness and psychological problems [29]. In contemporary China, some grandparents sincerely seek to provide care, guidance, and supervision for LBCs in a manner that decreases their chance of delinquent behavior [18,40,41], a finding in contrast to earlier studies arguing that LBCs usually exhibit delinquent behavior when grandparents are the primary caregivers [21,30]. However, most rural grandparents possess low academic qualifications and an outdated view of modern society; this prevents them from providing the proper care, supervision, and support LBCs seek in their academic, social and personal adjustment [21,30,33]. Even worse, in addition to grandparents’ conservative childrearing, their age (most grandparents caring for LBC are over 60 years old) places physical and psychological pressure on LBC, given that they are required to handle a substantial number of home and agricultural duties, which consume their study and leisure time, making the children feel marginalized compared with their peers [21,30,41]. Thus, the health status of the grandparents affects the decision-making process of migrant parents being both- and single-parent migrants to the cities. When it is good, it contributes to adequate caregiving and encourages migrant parents to save and continue to stay away [41]. The best-case scenario is when the grandparents can obtain a remittance from the migrant parents sufficient to relieve them from their farming burdens to allocate more time to support the health and welfare of their LBC, and thus improve their academic performance [18,41].

### 1.4. LBC and COVID-19 in China

The outbreak of the COVID-19 virus has led to severe global changes in social, political, and family settings; in particular, Chinese society has suffered from psychological, physical, and social problems arising from its early exposure to the pandemic [20,42,43,44]. Due to pandemic-related complications during the outbreak’s preliminary stage, parents and children were expected to quarantine at home [14,23,32,45]. LBC and migrant parents were already among the most disadvantaged groups in society, due to their long history of physical distance. The COVID-19 outbreak-related restrictions on people’s movements increased their difficulties [16,46]. Children who were already economically disadvantaged, such as LBC consequences of the pandemic regulation, restricted offline schooling, social interaction, and family earnings [20,42,45,47]. Explicitly, although the Chinese government has implemented several strategies to help the most vulnerable children to ensure their health, welfare and education, children’s mental health has suffered due to inequality of access and resource utilization [14,20,45,47]. Meanwhile, COVID-19 could force LBCs to lose the chance to play outside with their peers or to attend school, with consequences such as severe mental stress, anxiety, and depression [14,20,23,43,45,46,48]. In addition, migrant workers in China face particularly severe problems and challenges under the pandemic restrictions, and their unstable employment, income, and life security circumstances have worsened [24]. Most migrant parents became stuck in their home villages after returning for Chinese spring festival celebrations, and another group became trapped in their workplaces, and could not return to their homes [16,24,49,50]. A two-province survey study by Wang et al. assessed LBC; rural children who lived with both parents showed lower loneliness during the COVID-19 outbreak [23,50].

In contrast, during the national quarantine period, non-LBC could obtain more consideration and focus from their parents, which supported the parent-child relationship [23,44]. During the preliminary COVID-19 outbreak, LBC online learning motivation and engagement were primarily poor, as caregiver grandparents found it difficult to support and guide online learning via electronic devices, and follow-up by migrant parents was complicated by the pandemic [23,49]. Subsequently, due to poor communication between migrant parents and LBC regarding controlling the pandemic, the children faced a substantial fear of COVID-19, depression, stress, and anxiety [16,32,43,45,46,49,50]. Meta-analysis evidence indicated that LBCs were the most disadvantaged group of vulnerable children in China, as no one near them could perceive abnormal behavior or countermeasures [47]. However, little is known regarding the psychological, academic and parental attachment experiences of LBC during the preliminary stage of the pandemic. Furthermore, a large body of evidence regarding the well-being of LBC before and during the COVID-19 outbreak was gathered using quantitative research methods, which cannot explain the studied phenomena in-depth. Therefore, this study seeks to obtain more robust findings that complement the statistically oriented qualitative studies.

## 2. Method

### 2.1. Participants

This study examines the psychological well-being, academic adjustment, and quality of parental attachment of LBC during COVID-19 based on LBC word of mouth. Therefore, we adopted a qualitative research method consisting of interviews with LBC (face-to-face) to collect the evidence for the study [4,24]. The study primarily targeted an anonymous rural province in southwestern China, where many migrant workers originate, as it is less developed and offers fewer job opportunities than other provinces [4,48]. The research ethics committee of Shenzhen University approved the research protocol and study guidelines (LL2020002). The fourth author has a solid and continuous academic relationship with school principals and teachers in the studied province, who helped locate participant LBCs from different grade levels. Thus, we adopted the purposive sampling method to intentionally recruit potential and voluntary children’s participants in primary and junior secondary public schools from grade levels 4 to 8 with one parent or both parents who had migrated at least six months earlier to another province to seek better employment, and who were physically separated from their migrant parents (with single or both parents) during or after the COVID-19 pandemic. Subsequently, the legal guardians or caregivers of all children participating in our study signed an informed consent form. Eventually, the study acquired 22 voluntary LBC participants aged 10 to 15 years and a proportionate gender ratio (i.e., 11 boys and 11 girls).

### 2.2. Data-Gathering Instrument

The semi-structured face-to-face interviews were conducted with the LBC sample to reveal their experience during the pandemic (See Appendix A). The interview was conducted in early May 2021 by the corresponding author of the study, a Chinese native academic, and psychologist from the intended province. He has Chinese language proficiency with specific area dialects, and has expertise in counseling children with difficulties. At the same time, the second and third authors made audio recordings of the interviews, and took notes regarding the non-verbal expressions of the participants during the interviews. Later, with the assistance of a qualified translator, we transferred the interviewee’s responses from the Chinese (Mandarin) language to English. To ensure the interview findings’ trustworthiness, probing questioning tactics were adopted to provide the participants’ accounts’ consistency and accuracy. At the beginning of the interviews, we asked the participants to introduce themselves and be relaxed with us, in order to develop trust.

### 2.3. Data Analysis Procedures

This qualitative study is guided by predetermined themes that enabled us to adopt deductive means of thematic analysis. This method of data analysis allows the researcher to analyze the interview data according to the expected finding categories that emerged from the literature review. The study intended to analyze the data acquired through the semi-structured interviews by relying on academic adjustment, psychological well-being, and the quality of parental attachment of the participating LBC during the preliminary period of COVID-19. However, before analyzing the original transcript (in Mandarin) of the data that the first author transcribed, the transcript was professionally translated into English so that the second and third authors could perform the analysis. After the translation, the study of the transcribed data and the careful interpretation of these data led us to the three predetermined themes. Table 1 below presents the interview’s preliminary findings, which showed that five of the 22 participating LBC participants claimed their parents were divorced. As we expected, except for one child, we noticed that all participating children were cared for by grandparents [20]. In contrast, astonishingly, one child’s legal guardian was a school teacher who was not a relative. The child’s father was a migrant, and the mother was nowhere to be found.

### 2.4. Research Findings

#### 2.4.1. COVID-19 Outbreak and the Psychological State of LBCs

Concerning the rural parents’ migration, the size of remittance is an essential matter for left-behind children and their migrant parents. Most participants confirmed that their parents’ motives for moving to an urban locality were to meet their family’s obligations [49]. The psychological state of LBC is further subdivided into LBC’s view of COVID and LBC’s parents during COVID.

#### 2.4.2. LBC’s View of COVID

Due to parental migration, these LBCs underwent emotions of anxiety and sadness from their parents’ physical presence; with these feelings and emotional states, these LBCs perceived COVID-19 (See Table 2). Respondents shared fascinating psychological facts regarding COVID-19 that they came across. R2 stated that:

“COVID was a tiresome experience for me, lots of work and work, nothing else. It was a disease, but I was not scared of it. Sitting at home all the time and looking after my grandparents and young brother… my grandparents are very old… no sports, no meeting and playing with friends, and no excitement, just boring”.

While discussing their experiences, our respondents mentioned that they were not afraid of COVID-19. Notably, we observed that concerning COVID-19, not a single respondent exposed a view of panic and stress. R4, for example, stated that “there is no change in my life… all is the same as before, COVID brought no change for me… My sister used to bully me, and I will keep beating her… I help my grandmother with household chores such as cleaning and cooking, but after the pandemic, I do not want to help her with housework. I want to play and enjoy with my friends”.

Similarly, R7 argued that “this year my parents did not visit me; this was the only change which COVID brought me. Otherwise, all is the same as before… just studying and sitting at home is boring for me… COVID did not give me fear… I hope my parents will visit me soon and we have a good time together”.

Respondent R22 shared their experience of solitude and apathy during COVID-19 while staying at home, and revealed that “… I feel alone during a pandemic at home with grandparents only, no playing, no families, no excitement, and no friends… even in my studies there was no one who can guide me… I missed my school fellows and friends… and our playing time. I feel exhausted from housework and lazy staying at home and do household chores… but I am not scared of COVID-19. It is tiresome, nothing else… for me, it is scarier is to stay at home and work all time and stay away from my friends and games”.

While sharing the psychological state and experiences of LBCs, we understand that LBCs deal with pandemics with bravery, and are not bothered. Similarly, respondents R5 and R7 shared the same feelings of no fear and anxiety regarding COVID. Still, they revealed that the pandemic was unexciting; primarily, staying at home and staying away from friends and playing made them feel stressed. Coding the same facts, one of the respondents, R10, marked that “it was a boring time for me, but I do not feel any kind of fear from COVID”.

#### 2.4.3. LBCs’ View of Their Parents

Evidence from literature revealed that parent-child communication is crucial for the emotional well-being and pleasure of LBC. Furthermore, unstable parent-child interaction causes feelings of stress. Since most LBCs were living with their older grandparents, as a result of grandparents aging and inadequate emotional attachment, LBCs develop depression and sad feelings (See Table 3). Such a respondent, R5, revealed that:

“… My friends are happy, they eat together and play with their parents, and they do not have any issue with their studies… because their parents teach them and help them to read, I feel sad, and I wish that if my parents were with me, I would be happy like my friends, and I would be good in my studies, but my parents are not around… from COVID, I do not have any fear; I want my parents back, and I play and spend time with them… I know my mother is working to arrange school fees for my sister”.

LBCs explained their parents’ attitudes during COVID and shared their experiences with us. As respondent R4 explained, “my mother used to call us once a week, but she is busy with her work… for me, nothing changes, her behavior is the same; she calls as she used to do, I want her to come back and stay with me… my grandma mostly guides me to work hard for my studies”.

Furthermore, explaining the pandemic stress, respondent R10 outlined that “my parents used to visit us during Chinese New Year each year… but during COVID they used to do video call and sometimes a telephone call to ask about my grandparents and me… I was looking after myself during the COVID as per their advice, such as washing hands and not playing outside, wearing masks, and staying at home… but I think there is no change; all is the same for me”.

While sharing their experience, respondent R1 revealed, “My parents used to visit us once a year, mostly on Chinese New Year… but they used to talk to us every day… sometimes, like when my big brother broke his legs, then they paid us a quick visit to solve his health issues, and even last time they paid lots of money for his treatment and stayed with us until he recovered… whenever something happened, they used to visit, such as once my brother fought with someone and broke his leg, and my parents visited us—otherwise they visit at the new year. Apart from that, they used to visit due to my brother’s behavior”.

While discussing their diverse experiences during COVID, LBCs shared their parental advice to stay safe from COVID and their parental communication and their personal feelings toward the pandemic, and revealed that they were not afraid of COVID-19. Notably, most of them talk about the household obligations and tiresome routines that make them more occupied, and few of them revealed that due to this physical fatigue, they felt tired and depressed.

#### 2.4.4. COVID-19 and LBC Academic Adjustment

During the pandemic outbreak, the Ministry of Education in China launched an online schooling system to prevent harm to education, and successfully launched online classes initiatives during COVID-19. Nevertheless, the sudden outbreak significantly hindered the Ministry of Education; providing visual aids to rural school children was a challenge, since rural children have limited access to the latest technology gadgets. Shortly, with the effective recommendation for using all available visual aids such as TV, WeChat, and DingTalk, we will be able to execute the online learning process smoothly. Consequently, the ministry accomplished the launch of online classes in rural areas. Our respondents shared how they undergo online schooling experiences, their household obligations, and how these influence their online learning with us. The academics of LBCs are presented under these subthemes as LBC’s view of school and LBC’s and caretakers’ responsibilities (See Table 4).

Discussing schooling experiences, R2 stated the following: “… Our school started online classes for us during the pandemic period… but most of the time I do not understand my teacher, I feel alone and bored sitting and waiting for the teacher to ask me some questions, and sometimes she did not ask any question… I cannot see my friends; I cannot talk to my teacher… a few days I do not understand what she is teaching and which subject… like audio tape recordings I have to listen to, because they’re at home, no one is able to help me in my studies”. Regarding the online classes, all respondents mentioned the experience of the lifelessness of online studies, and reported this experience as a learning constraint.

Some respondents revealed the swiftness of online classes, and the respondent marked as R4 stated the following: “I believe online classes are faster than real classes, because teachers only speak and I do not have any chance to participate in class, just listening… due to teachers’ speed of talking, sometimes I do not understand anything, and I cannot stop her to repeat for me… besides, sometimes due to housework if I am late for the class then it becomes difficult for me to understand the teacher… my grandparents are old, so I have to do more household chores and have less time left for my studies… I struggle with my studies because no one can teach me at home”.

Respondents mentioned the physical absence of their classmates and friends and missed their class activities. R18 reported the following: “It is boring to sit and watch TV to understand the teacher’s words. The pace of teachers is mostly very fast. I do not understand sometimes… and if I skip some parts, then no friend or classmate can help me with what she is teaching and which subject. It is very boring just listening to the teacher, no friends, no fun, just words; I miss my class, my activities, and their help”.

When comparing online classes with offline classes, one respondent, R20, reported: “Online classes are good to enable us to study at home, but online classes are not interesting, such as my school classes where all my friends were together… I can ask the teacher what I do not understand, and I missed my classes and friends; online classes are just listening and listening”.

Respondent 8 shared her online class experiences with her mother during COVID-19, and revealed that “my mother helped me a lot to learn from online classes; she teaches me how to take notes and not just listen to the teacher, and how to use the internet to read more about English, because my English is weak compared to Mathematics and Chinese… since she knows my weak subject she supported me to read more English, and I am happy I scored good grades in English”.

Another respondent, R16, shared the parental help regarding his studies, and pointed out: “I did not feel any change during COVID-19, I feel brave and strong… my parents helped me in my studies by teaching me through video calls whatever I do not understand from an online class, which I discuss with my parents to help me; I enjoyed the online classes, and they were a nice experience”.

Moreover, R13 stated the motive of studying hard, and revealed: “I know my parents are working hard for me, and I have to study hard to achieve excellent grades to have a better life for them… I achieved high grades in class”.

Discussing parents’ expectations to achieve high academic grades, respondent R15 added, “I feel difficulty in English; parents always guide me to work hard to achieve the high grades and read more books… but there is no one who can help me to read English books and stories, I wish my father was there to help me and read for me to understand like my friends… but he is so busy with work and had no time even for the call”.

#### 2.4.5. LBC and Their Legal Guardian’s Caregiving

Most of the respondents were under the care of their grandparents, and shared their academic experiences and caretaking obligations, and revealed different facts, as R2 added: “I helped my grandparents in the household by cleaning, cooking, and washing, because they are farmers and they feel tired, so I helped them, but sometimes I feel so tired to do household chores, and I cannot pay attention to my studies and class work”.

Likewise, R6 revealed the following: “During COVID-19, my parents were in Guangzhou for their work, and mostly my parents called me weekly, and most commonly asked about grandparents and my health… they used to advise me do not go out and stay at home and look after grandparents help them in households and study hard; this is their most common advice for me whenever they used to call us… besides, there is no change during COVID, and only little new advice, such as washing hands and not playing with friends”.

Almost all respondents mentioned and discussed the influence of health and aging issues of their grandparents on their academic learning process, and R22 added the following: “Most of the time, I have to manage the household and help my grandma in cooking, cleaning, and washing dishes because she is blind, so I helped her; grandpa is a farmer, and he used to work in the field and sometimes he needs my help too because he is too old to work alone… but all this work made me feel so tired and weak… during the online classes sometimes I cannot focus on my studies, and due to my tiredness I cannot pay attention to online classes. Sometimes I try to understand mathematics because my mathematics is very weak, but no one is there to teach me at home and no friends can guide me to study”.

Notably, regarding online classes, all respondents mentioned several facts that cause hindrance in their academic learning processes, such as physical fatigue and household obligations, grandparents’ poor health status, online classes’ teaching style, and a lack of guidance and help from parents and grandparents. As a result, this adds up to more worries for these LBCs (See Table 5).

#### 2.4.6. LBCs’ Parental Attachment during COVID-19

Most LBCs share their parental relationships and attachments during the pandemic through social media, as well as the advice they receive. For example, respondent R22 disclosed his academic and COVID precautions, parental guidance, and its influence on his lifestyle (See Table 6).

Further, respondent R2 added the following: “… my brother and I were living with grandparents, and we looked after them. My parents talk to us once a day as usual… I think our life had already changed before COVID when they left us… when I see myself and my friends, we have a big difference. They have family we do not have. Nothing else. I do not have any wish from my parents, and I do not need anything”.

Respondent R4 states: “When I was born, and my father was in jail, my mother left my two elder sisters and me with our grandparents to make money for us. It’s been six years now. During COVID, she was busy with her work, and she used to call very little to ask about us… Her big words for us were not to go out, and to stay at home to help our grandparents. Grandma is sick and cannot cook, so I clean, my sisters cook, and I help with the farming too, I think my mother loves us… that is why she works very hard”.

Respondent R5 remarked the following: “My mother was calling me and was asking about my health when she gets time. My stepfather hates me, so she cannot stay with me since I was with my grandparents forever during the pandemic. I was so sad and depressed when I saw my friends with their families and having fun. I am not lucky like them”.

Likewise, respondent R7 remarked: “Actually, I do not feel fear from COVID. My parents used to call me daily and guide me on how to be safe and healthy, like staying at home and helping my grandparents. When I was a child, I wished my parents would return and spend more time with me… I know my parents are working hard for me to provide me with milk and food, and they promised to buy me a new computer to study”.

Similarly, R18 mentioned her father’s instructions, and added: “My father loves me a lot. He always tries to be around me and calls me three times a day. During COVID, I was with my teacher. After my mother left me, my teacher looked after me… my dad told me not to eat indiscriminately outside, to buy as much food as possible when I was at home, to not starve myself, to study and listen to my teacher, and to not go out at night”. 

Respondent R20 adds the following: “My father is working hard for us. He is a chef and used to call me frequently, but I was so bored. I was not afraid of COVID-19. It makes no difference to me. All is the same. It’s boring at home, and I cannot go out to play. Dad called daily, saying the same: don’t go out, stay home, and be safe. My grandparents gave me money to buy food online”.

Furthermore, respondent R14 mentioned: “My parents are separate from each other. My father works in Liangjiang (city), and my mother works in Guangzhou. My father didn’t call me, even during COVID-19. He didn’t call, and my mother was very busy, but when she used to call me, she fought with me. I wish she didn’t call me and stayed busy. My mother has to take care of my sister because she needs a lot of tuition for her third year of senior high school”.

Likewise, R16 added: “Our parents left us the last two years, and during COVID, they were not with us. Before COVID-19, my mother visited us on birthdays and Chinese New Year. The pandemic was serious, so they were in contact through calls. Whenever they got the time, they used to call us and ask about our health, but fought with us sometimes… I know my parents left us for money and to make us eat good, sometimes I wish I had good food such as milk, snacks and good times with my parents… most of the time when my mother used to call me, she fights with us and stays angry, because her job is difficult. She is tired of it, and during COVID, they have not had a job… sometimes I think this is why they left us”.

## 3. Discussion

The COVID-19 outbreak has caused extensive lifestyle changes worldwide. These sudden changes have injured the well-being of vulnerable children, particularly LBCs in rural China, who experienced the adverse effects of the pandemic earlier and have been affected more intensely than anyone else. However, few quantitative studies have addressed the well-being of LBC during the early phase of the COVID-19 outbreak, and those that have are insufficient in describing the experience. Thus, this study represents a rare attempt to explore the well-being of LBC in China during COVID-19 using a face-to-face interview to assess psychological and academic effects and parent-child attachment. Based on the findings of the analyses, we discuss the following three themes: the psychological and educational well-being, and the parent-child extension of LBC during the pandemic.

### 3.1. Psychological Adjustment of LBC

Our interviews with LBCs revealed that during the initial and later stages of the pandemic, the children exhibited considerable difficulty adjusting their emotional and psychological state owing to the radical lifestyle changes caused by the COVID-19 pandemic [23,25]. The lockdown and quarantine regulations imposed by the government restricted social interaction and movement, diminishing the ability of migrant parents to visit their children in their home villages during critical the Chinese New Year period, and fostering a negative psychological state, including feelings such as loneliness, depression, anxiety, and sadness, as witnessed in face-to-face interviews [7,23,27,28,50]. A possible explanation for this finding is that, unlike ordinary children, LBC felt considerably disheartened and annoyed because COVID-19 spoiled a rare opportunity to meet their parents during the national holiday. Additionally, social isolation decreased schooling time, as well as peer interaction, which represents a rare chance for LBCs to play, communicate and socialize. Our results indicate that none of our participants’ LBCs experienced a significant fear of COVID-19 during the early stage of the outbreak; the possible explanation for this might be that these children have only a minor threat of COVID-19 as compared to their adults. This finding contradicts studies that found LBCs to display excessive fear of the pandemic [16,32,43,45]. One potential reason for this unexpected finding is that LBCs have a history of suicidal ideation and suicide attempts, given their unhappy and often hopeless lives [21,22,30,31,32]; thus, LBCs might not fear a dangerous disease. Another explanation might be that the local health and welfare authority or the migrant parents failed to inform caregiving grandparents of the gravity of the situation, and how to take proper preventative measures. In any case, having no fear of the pandemic cannot make things better. Instead, one becomes a hazard to both oneself and the family members one lives with.

Furthermore, our findings revealed that most LBCs feel they do not need their parents, which reflects their long record of hopelessness and agony regarding the absence of their parents. Previous research has found that LBCs are always excited and wish to see their migrant parents [7,26,28]. A study finding revealed that extreme isolation from their parents and friends during COVID-19 make these LBCs experience more depression, making some claim that they do not need parents. LBCs having no fear of COVID-19 was the surprising finding of the study.

### 3.2. Academic Adjustment of LBC

An extensive number of studies claim that LBC academic performance is significantly lower than that of their counterparts, as the long working hours of migrant parents limit the time they have available to help their children with their school work, and because caregiving grandparents are insufficiently educated to help [1,3,8,9,10,12]. During the pandemic, LBC academic adjustment worsened because school was conducted through an online platform, requiring strict parental guidance. LBC academic engagement and motivation were notably jeopardized during the pandemic, as these children often do not attend class regularly as a result of irregular internet access [14,20,45,47] and intensive household and agricultural duties imposed by their grandparents [7,9,11,12,25,30,37,41]. Although parents migrate to cities to obtain an income adequate to the needs of the rural village household and the academic needs of their LBC, there is no assurance that they can earn enough [7,37]. Consequently, our participants stated that they had substantial difficulty attending their online classes due to physical and psychological fatigue caused by their daily duties—duties arising from the inability of their migrant parents to send home enough money to fully cover household needs. Our findings also reveal that the online teaching method was not interesting enough to stimulate all students and promote discussion among peers. Accordingly, in addition to the limited educational background of the caregivers, the insufficient peer and parental support worsened the academic adjustment of LBCs during the pandemic. In sum, our qualitative analytical findings concerning LBCs’ intellectual well-being during the COVID-19 outbreak indicate that a lack of parental support, learning resources and disorganized online teaching methods ultimately lead to poor academic performance.

### 3.3. Parental Attachment and LBC during COVID-19

This study demonstrates that a long parental absence from a rural household negatively influences the relationships between the parent and child. This phenomenon was evident during the pandemic, when children needed their parents like never before. Plausibly, our study findings reveal that parent-child attachment predicts the psychological and academic well-being of LBCs [3,35,38]. During the preliminary stage of the pandemic, a small number of migrant parents reached out to their LBCs through video and voice calls to provide information on how to prevent COVID-19. However, not leaving the home and helping their grandparents were systematic instructions that resulted in unnecessary perceptions regarding the pandemic. Regrettably, few LBCs realize and understand that their only occasional communication and reduced attachment with their migrant parents resulted from the desire of their parents to help them [5,18,19]; instead, they were subjected to alarming psychological anguish. Concurrently, during the early stage of the outbreak, most migrant parents lost their jobs or had their income reduced [23,24,50], which prevented them from visiting their LBC, made it impossible to send adequate remittances, and frequently made long-distance communication more difficult. Thus, one can claim that the direct adverse effect of COVID-19 on the income or job security of migrant parents endangered the academic and psychological well-being of their children—a problem that urgently demands attention.

## 4. Conclusions

Our study is the first to directly interview left-behind rural children in China who have experienced the COVID-19 pandemic before any other vulnerable children worldwide. Hence, the present study elucidates that LBCs claimed they no longer need their parents, representing their extreme loneliness and depression. Moreover, the social isolation took away the chance to meet their migrating parents and school peers, and engage in outdoor activities, leading them to stay at home with their grandparents, who were old enough to be unhelpful in their academic, psychological and physical well-being.

### Implications and Limitations

In this qualitative study, which enabled us to examine the LBC experience during the preliminary stage of the pandemic, we sought to emphasize critical practical implications for China and other countries still involved in the fight against COVID-19. In light of our study findings, public health and welfare agencies—including governmental and non-governmental institutions—should carefully consider providing aid to vulnerable children, such as LBC. They are substantially more disadvantaged than other children. Our results also suggest that village or district health centers play a central role by better addressing their services, such as counseling and awareness regarding COVID-19, nutrition and healthy well-being, to rural societies that find it challenging to access digital-age information networks. To enhance the effectiveness of online classes, school leaders and teachers should develop stimulating online teaching methods that help students feel they are sitting in the classroom among their peers, and design strategies for LBCs and other unprivileged children to obtain extra support from their teachers and peers.

Our qualitative study performed in province Y has two main limitations. The first is that because the study uses a qualitative, interview-based approach, its findings cannot be generalized to other provinces. This limitation should encourage future researchers in and outside China to pursue unusual qualitative studies, which can provide a deep understanding of LBC. Second, this interview study considers only LBC, whereas it might also have addressed migrant parents, legal guardians, teachers, and community service personnel. This limitation should be regarded by scholars who wish to explore this global phenomenon from different individual perspectives, which could provide a holistic understanding—a goal most studies neglect.

## Figures and Tables

**Table 1 children-09-01317-t001:** LBC Basic description.

Respondent	Gender	Age	Grade Level	Parent Migrant Duration	Caretaker of LBC during COVID-19	LBC Phrases for COVID-19	LBCs and Migrant Parents’ Relationships during COVID-19
R1	Male	11	4	10 years	GP	Not afraid	Video/audio communication once a day
R2	Male	15	8	7 years	GP	Not afraid	Video/audio communication once a day
R3	Female	13	7	No idea	GP	Not afraid	Rare communication
R4	Male	14	7	6 years	GP	Not afraid	Rare communication
R5	Male	14	7	3 years	GP	Not afraid	Video/audio communication once a day
R6	Female	14	7	10 years	GP	Not afraid	Rare communication
R7	Female	13	7	4 years	GP	Not afraid	Weekly communication
R8	Female	12	6	3 years	GP	Not afraid	Video/audio communication once a day
R9	Female	12	6	2 years	GP	Not afraid	Video/audio communication once a day
R10	Female	12	6	2 years	GP	Not afraid	Video/audio communication once a day
R11	Female	10	4	6 years	GP	Not afraid	Twice or three times call in a week
R12	Male	11	5	2 years	GP	Not afraid	Video/audio communication once a day
R13	Male	12	5	5 years	GP	Not afraid	Once a week and sometimes after two weeks
R14	Male	10	4	3 years	GP	Not afraid	Call frequently mostly the father, not the mother she is busy she calls on Sunday
R15	Female	10	4	No idea	GP	Not afraid	uncertain very little once in a week
R16	Female	10	4	2 years	GP	Not afraid	Video/audio communication once a day
R17	Male	11	4	5 years	GP	Not afraid	Video/audio communication once a day
R18	Female	14	7	10 years	ST	Not afraid	Video/audio communication once a day
R19	Male	14	8	3 years	GP	Not afraid	Video/audio communication once a day
R20	Male	14	7	8 years	GP	Not afraid	Video/audio communication once a day
R21	Female	13	7	2 years	GP	Not afraid	Weekly communication
R22	Male	15	8	10 years	GP	Not afraid	Weekly communication

Note: Grandparents “GP” and School teacher “ST”.

**Table 2 children-09-01317-t002:** LBCs’ view of COVID.

Respondent	Siblings	Class Level	LBCs’ Household Activities	LBCs’ Feelings during COVID
R2	No	6	Laundering and cleaning	Unhappy and boring
R4	2 sis	8	Laundering and helping grandparents in farm work	Depressing and unhappy
R7	2 bro	6	Kitchen work and cleaning	Sad
R22	1 bro	7	Clean up the house and farm work	Sorrowful
R8	1 bro	6	Domestic obligations and farm work	Boring and stressful
R10	1 bro	6	Laundering, cleaning, and helping in farm work	Discontented from friends
R20	1	8	Laundering, cooking and farming	Joyless

**Table 3 children-09-01317-t003:** LBCs’ view of their migrant parents.

Respondent	Class Level	LBCs’ Feelings for Their Parents	COVID and LBCs’ Parental Guidance
R1	6	Come back and stay with us	Stay at home and help grandparents
R4	8	It’s ok whether they are with me or not	Stay at home
R7	6	I do not want them anymore	Do not play around and focus on the study
R19	7	I know they are busy	Do not play around and focus on the study
R5	6	I wish I could have stayed with them like my friends	Keep washing your hands
R10	6	Come back and stay with me	Wear a mask and avoid friends and people
R11	8	It’s ok they are with me or not	Stay safe and help grandparents with housework

**Table 4 children-09-01317-t004:** LBC academic adjustment during COVID-19.

Respondent	Grade	Weak in Subject	Online Classes Feedback
R2	8	English and Mathematics	Uninteresting
R4	7	Mathematics	Tiresome
R8	6	Mathematics	Tiresome and unexciting
R13	5	English and Mathematics	Boring and complicated
R15	4	English	Very complicated
R16	4	English	Difficult to understand and unexciting
R18	7	I do not have any kind of trouble	Unexciting
R20	7	Good in my studies	Tiresome
R22	8	Mathematics	Tiresome

**Table 5 children-09-01317-t005:** LBC and their legal guardians’ caregiving.

Respondent	COVID and Grandparents’ Caretaking	Health Status of Grandparents	Reason LBCs Revealed for Parents Leaving
R9	Grandma	Very old and cannot walk properly	Lower-income
R18	Myself	Old enough	Lower-income and family needs
R5	I rely more on my mother, and my father doesn’t care much about me	Sick and old	Lower-income
R14	No one cared for me	Very sick and old	Lower-income and family needs
R16	Grandma and dad	Very sick and old	Money for us
R20	Myself	Old enough, so I do household chores	Lower-income
R22	Myself	Grandma is blind, and grandpa is old	Lower-income

**Table 6 children-09-01317-t006:** LBCs’ parental attachment during COVID-19.

Respondent	Age	Parents Migration Years	LBCs’ Feelings for Their Parents	Hopes from Parents
R4	14	7 years	My parents do not love me; they want to make money	Nothing I want from them
R5	14	8 years	My mother loves me; my father doesn’t care much about me	I do not need anything
R7	13	3 years	No-one loves me	Come back and live with us
R14	10	5 years	I do not know	Come back and spend time with me
R16	10	4 years	Grandma loves me	I do not need anything
R20	14	8 years	No-one loves me	There is nothing I want

## Data Availability

The datasets generated and analyzed during the current study are available from the corresponding author on reasonable request.

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
