# Peer review of "COVID-19 and Vulnerable Children Well-Being: Interview with Left-Behind Children in Rural China"

_children, 2022, doi:10.3390/children9091317_

Round 1
Reviewer 1 Report
This article regards a qualitative study that explored the psychological, academic, and parental attachment experiences of rural LBC during COVID-19.
The article is well written. To improve the article I have some questions:
-Authors should better describe the semistructured interview used in their study (How were the questions created? Did an earlier basis already exist or were they made ad hoc?).
-Who has been asked for informed consent to conduct interviews with children?
-Among the participants the children are all looked after by grandparents, only one by the teacher, how was this variable checked?
-The results are a bit 'confused, maybe it would be useful to make a table where to report the issues emerged form semistructured interview
-When Authors say: “Most of respondents..” they should specify the number of respondents.
-The authors have analyzed gender differences, are there?
Author Response
August 03, 2022
Children (ISSN 2227-9067)
Dear Editors and reviewers,
I am writing this letter on behalf of my coauthor regarding our manuscript Ref: Submission ID children-1854945, entitled "COVID-19 and Vulnerable children well-being: Interview with left-behind children in rural China". We want to express our appreciation to the respected editors and reviewers for providing us the constructive comments and suggestions for the manuscript to be considered for publication with minor adjustments. As you can see below, we responded to the first reviewer's inquiries before the manuscript was published, and we would like to stress that most of the suggestions the reviewers gave were identical, so we tried to respond to them as one to avoid redundancy.
Reviewer 1 Comments
Comment 1: Authors should better describe the semi-structured interview used in their study (How were the questions created? Did an earlier basis already exist, or were they made ad hoc?).
Author's Response: Thank you for the genuine suggestion regarding our interview protocol. The interview questions were designed from the input we gathered from recent literature that explores children's well-being during the Covid-19 pandemic. Likewise, the authors (native Chinese) played a significant role in supplementing crucial interview questions that would make the study worthier. We will try to highlight these points in the manuscript.
Comment 2: Who has been asked for informed consent to conduct interviews with children?
Author's Response: Due to the nature of this study participants, we were required to meet their homestay parent or legal guardians (e.g., grandparents or relatives).
Comment 3: -Among the participants, the children are all looked after by grandparents, only one by the teacher; how was this variable checked?
Author's Response: If we understand reviewer 1 question. During our interview, we found out that one girl participant was under her school teacher's custody since her mother was missed and her father was working in the city. Unfortunately, the girl has no relatives or grandparents who can look after her.
Comment 4: -The results are a bit 'confused; maybe it would be helpful to make a table where to report the issues that emerged from semistructured interview.
Authors' Responses: We will try to manage it.
Comments 5: -When Authors say: "Most of respondents.." they should specify the number of respondents.
Author's Response: Well noted.
Comment 6: -The authors have analyzed gender differences, are there?
Author's Response: We have already addressed the participant section and table 1.
Reviewer 2 Report
Dear Authors,
Thank you for the opportunity to review the manuscript “ My life is more than COVID-19: Interview with left-behind children in rural China”.
I make the following recommendations to improve the manuscript:
1) TITLE: Please think about concise, specific and relevant title.
2) ABSTRACT:
a. The abstract should be a total of about 200 words maximum.
b. The abstract should be an objective representation of the article.
c. Place the question addressed in a broad context and highlight the purpose of the study. What is the most important aim of research?
d. Results should summarize the article's main findings. Please show the most important conclusion.
e. Abbreviations should be defined the first time.
3) INTRODUCTION:
a. The introduction should briefly place the study in a broad context and highlight why it is important. The current state of the research field should be reviewed carefully and key publications cited.
b. Please change, reference numbers should be placed in square brackets [ ],
c. The introduction should be short, concise. Please write the most important information about psychological challenges of left-behind children and influence of COVID-19 of LBC.
d. At the end of the introduction, you should mention the main aim of the work.
e. Please Use the Microsoft Word Template to prepare manuscirpt.
4) MATERIALS AND METHODS:
a. It seems necessary to clarify how the representative sample was calculated and to make a flowchart of the enrollment of respondents and study procedures. What are the inclusion/ exlusion criteria?
b. How did you collect data about socioeconomic variables/ demographic variables? Did you use Self-Authorship Survey or standardized survey instruments?
5) RESULTS : Where is section: results?
6) DISSCUSION:
a. Each of the results should be mentioned in more depth, indicating references.
b. I think the authors should compare the results obtained with another research that analysis of left-behind children. Each of the results should be mentioned in more depth, indicating references. Please remember about your aim.
c. I think the authors should compare the results obtained not only with similar studies during the pandemic but also with previous studies, to establish a comparison between psychological/ academic challenges of LBC before and after the COVID-19 pandemic.
7) CONCLUSION: Where are conclusions?
8) REFERENCE:
a. I suggest conducting a new literature review. This article needs the newest references (written after 2020).
b. Please check the Instructions for Authors. References should be described as follows, depending on the type of work: https://www.mdpi.com/journal/ijerph/instructions.
Author Response
August 03, 2022
Children (ISSN 2227-9067)
Dear Editors and reviewers,
I am writing this letter on behalf of my coauthor regarding our manuscript Ref: Submission ID children-1854945, entitled "COVID-19 and Vulnerable children well-being: Interview with left-behind children in rural China". We want to express our appreciation to the respected editors and reviewers for providing us the constructive comments and suggestions for the manuscript to be considered for publication with minor adjustments. As you can see below, we responded to the first reviewer's inquiries before the manuscript was published, and we would like to stress that most of the suggestions the reviewers gave were identical, so we tried to respond to them as one to avoid redundancy.
Reviewer 2 Comments
1) TITLE: Please think about concise, specific and relevant title.
Author's Response: The suggestion is considered, and I revised the title.
2) ABSTRACT:
- The abstract should be a total of about 200 words maximum.
- The abstract should be an objective representation of the article.
- Place the question addressed in a broad context and highlight the purpose of the study. What is the most important aim of research?
- Results should summarize the article's main findings. Please show the most important conclusion.
- Abbreviations should be defined the first time.
Author's Response: We are pleased reviewer two has guided us to polish the abstract. In line with the suggestion, necessary actions are taken.
3) INTRODUCTION:
- The introduction should briefly place the study in a broad context and highlight why it is important. The current state of the research field should be reviewed carefully and key publications cited.
- Please change, reference numbers should be placed in square brackets [ ],
- The introduction should be short, concise. Please write the most important information about psychological challenges of left-behind children and influence of COVID-19 of LBC.
- At the end of the introduction, you should mention the main aim of the work.
- Please Use the Microsoft Word Template to prepare manuscirpt.
Author's Response: The authors strived to make the manuscript as precise as possible, although the majority of the literature cannot be amended for the sake of the quality of the manuscript. Besides that, necessary adjustments are required to be made with extensive revision.
4) MATERIALS AND METHODS:
- It seems necessary to clarify how the representative sample was calculated and to make a flowchart of the enrollment of respondents and study procedures. What are the inclusion/ exlusion criteria?
- How did you collect data about socioeconomic variables/ demographic variables? Did you use Self-Authorship Survey or standardized survey instruments?
Authors' Responses: Due to reviewer 2, we have already addressed the inclusion or exclusion criteria to sample our participants. Regarding socioeconomic status (SES), since the research method is qualitative and SES is not the study variable, we haven't surveyed or measured this variable.
5) RESULTS : Where is section: results?
Author's Response: "Research Finding" is the result section. These terms are interchangeable in the article writing.
6) DISSCUSION:
- Each of the results should be mentioned in more depth, indicating references.
- I think the authors should compare the results obtained with another research that analysis of left-behind children. Each of the results should be mentioned in more depth, indicating references. Please remember about your aim.
- I think the authors should compare the results obtained not only with similar studies during the pandemic but also with previous studies, to establish a comparison between psychological/ academic challenges of LBC before and after the COVID-19 pandemic.
Author's Response: According to the suggestion from our respected reviewer 2, we have included recent references that would polish the manuscript and refine the discussion section.
7) CONCLUSION: Where are conclusions?
Author's Response: The conclusion section is included in the manuscript.
8) REFERENCE:
- I suggest conducting a new literature review. This article needs the latest references (written after 2020).
- Please check the Instructions for Authors. References should be described as follows, depending on the type of work: https://www.mdpi.com/journal/ijerph/instructions.
Author's Response: The comments are considered.
Round 2
Reviewer 1 Report
I thank Authors for responding to my comments and for adding some parts to article. Despite this, I still have some doubts regarding the method and results. Can Authors insert some sample questions of the interview conducted?
Regarding results, there is still confusion, an additional table with the main issues emerged during interviews could help the reader.
Author Response
August 15, 2022
Children (ISSN 2227-9067)
Dear Editors and reviewers,
I am writing this letter on behalf of my coauthor regarding our manuscript Ref: Submission ID children-1854945, entitled " COVID-19 and Vulnerable children well-being: Interview with left-behind children in rural China". We want to express our appreciation to the respected editors and reviewers for providing us the constructive comments and suggestions for the manuscript to be considered for publication with minor adjustments. As you can see below, we responded to the first reviewer's inquiries before the manuscript was published, and we would like to stress that most of the suggestions the reviewers gave were identical, so we tried to respond to them as one to avoid redundancy.
Reviewer 1 Comments
I thank Authors for responding to my comments and for adding some parts to article. Despite this, I still have some doubts regarding the method and results. Can Authors insert some sample questions of the interview conducted?
Author's Response: We have attached the interview outlines in the Appendix-A section. Kindly refer to the interview questions.
Regarding results, there is still confusion, an additional table with the main issues emerged during interviews could help the reader.
Authors' Response: To avoid any mix-up, the authors managed to knit the findings in (see Tables 2-4).
Reviewer 2 Report
Dear authors,
You can find my suggestions in order to improve your manuscript:
1) Introduction should be short and concise
2) I can't find information regarding:
a) how the representative sample was calculated and the scheme used to recruit responders and the experiment's procedure
b) inclusion/exclusion criteria
3. Results and discussion should be clearer
4. Please use these instructions for authors when citing references: https://www.mdpi.com/journal/ijerph/instructions.
Author Response
August 15, 2022
Children (ISSN 2227-9067)
Dear Editors and reviewers,
I am writing this letter on behalf of my coauthor regarding our manuscript Ref: Submission ID children-1854945, entitled " COVID-19 and Vulnerable children well-being: Interview with left-behind children in rural China". We want to express our appreciation to the respected editors and reviewers for providing us the constructive comments and suggestions for the manuscript to be considered for publication with minor adjustments. As you can see below, we responded to the first reviewer's inquiries before the manuscript was published, and we would like to stress that most of the suggestions the reviewers gave were identical, so we tried to respond to them as one to avoid redundancy.
Reviewer 2 comments
1) Introduction should be short and concise
Author's Response: It is well understood that the introduction section is a bit large, so we strived to precise it.
2) I can't find information regarding:
- a) how the representative sample was calculated and the scheme used to recruit responders and the experiment's procedure
Author's Response: Unlike quantitative research design, our study, which adopted a qualitative research design, doesn't require a probability sampling method that requires a rigorous sampling method to obtain a convenient participant representing the entire target group. Our study is qualitative and uses a non-probability sampling method; explicitly, this study adopted purposive sampling to select our participants according to the selection criteria.
- b) inclusion/exclusion criteria
Authors Response: We had already mentioned before that the study selected voluntary children's participants in primary and junior secondary public schools from grade levels 4 to 8 with one parent or both parents who had migrated at least six months earlier to another province to seek better employment and who were physically separate from their migrant parents (with single or both parents) during or after the COVID-19 pandemic. (See page 4, participants section)
- Results and discussion should be clearer
Author's Response: The authors strived to knit the finding and discussion in table form. Kindly see Tables 2 -4.